# Destabilization of the Charge Density Wave and the Absence of Superconductivity in ScV_6_Sn_6_ under High Pressures up to 11 GPa

**DOI:** 10.3390/ma15207372

**Published:** 2022-10-21

**Authors:** Xiaoxiao Zhang, Jun Hou, Wei Xia, Zhian Xu, Pengtao Yang, Anqi Wang, Ziyi Liu, Jie Shen, Hua Zhang, Xiaoli Dong, Yoshiya Uwatoko, Jianping Sun, Bosen Wang, Yanfeng Guo, Jinguang Cheng

**Affiliations:** 1Beijing National Laboratory for Condensed Matter Physics and Institute of Physics, Chinese Academy of Sciences, Beijing 100190, China; 2School of Physical Sciences, University of Chinese Academy of Sciences, Beijing 100190, China; 3School of Physical Science and Technology, Shanghai Tech University, Shanghai 201210, China; 4ShanghaiTech Laboratory for Topological Physics, Shanghai Tech University, Shanghai 201210, China; 5Songshan Lake Materials Laboratory, Dongguan 523808, China; 6Institute for Solid State Physics, University of Tokyo, Kashiwa 277-8581, Chiba, Japan

**Keywords:** ScV_6_Sn_6_, charge density wave, high pressure

## Abstract

*R*V_6_Sn_6_ (*R* = Sc, Y, or rare earth) is a new family of kagome metals that have a similar vanadium structural motif as *A*V_3_Sb_5_ (*A* = K, Rb, Cs) compounds. Unlike *A*V_3_Sb_5_, ScV_6_Sn_6_ is the only compound among the series of *R*V_6_Sn_6_ that displays a charge density wave (CDW) order at ambient pressure, yet it shows no superconductivity (SC) at low temperatures. Here, we perform a high-pressure transport study on the ScV_6_Sn_6_ single crystal to track the evolutions of the CDW transition and to explore possible SC. In contrast to *A*V_3_Sb_5_ compounds, the CDW order of ScV_6_Sn_6_ can be suppressed completely by a pressure of about 2.4 GPa, but no SC is detected down to 40 mK at 2.35 GPa and 1.5 K up to 11 GPa. Moreover, we observed that the resistivity anomaly around the CDW transition undergoes an obvious change at ~2.04 GPa before it vanishes completely. The present work highlights a distinct relationship between CDW and SC in ScV_6_Sn_6_ in comparison with the well-studied *A*V_3_Sb_5_.

## 1. Introduction

The kagome lattice, which is composed of a corner-sharing triangular network, has gained considerable theoretical interest due to the presence of Dirac points, van Hove singularities, and geometrically driven flat bands in a tight-binding model [1,2,3,4,5,6]. Depending on the band filling and pertinent interactions, various electronic states have been proposed theoretically, such as spin density wave [7], charge density wave (CDW) [8], bond order [9], and exotic superconductivity (SC) [3,10,11], etc. In contrast to the prevailing theoretical endeavor, experimental realizations of kagome metals hosting the above-mentioned rich electronic phenomena are scarce. Recent discoveries of vanadium(V)-based kagome metals, *A*V_3_Sb_5_ (*A* = K, Rb, Cs), have attracted considerable research interest, and they have now emerged as an important material platform to study the interplay among non-trivial band topology, CDW, nematic order, and SC [11,12,13,14,15,16,17,18,19,20,21,22,23,24,25,26,27]. Prior to the superconducting transition at low temperatures, i.e., *T*_c_ = 0.93–2.5 K, they undergo a second-order CDW transition at *T** = 78, 104 and 94 K for *A* = K, Rb, and Cs, respectively [11,14,15,16,17,18,20,21,28]. The nature of CDW and SC, as well as their interplay, has been the subject of extensive investigations in the last three years, resulting in many intriguing findings pertinent to the kagome lattice geometry, non-trivial band topology, and electron correlations in this system [6,8,10,11,12,13,14,15,16,17,18,19,24,25,26,28,29,30,31,32]. Among those studies, the high-pressure approach plays an important role in uncovering the intimate relationship between these intertwined orders [12,26,31,32]. For example, an unusual “M”-shaped double superconducting dome appears in CsV_3_Sb_5_, accompanying the monotonic suppression of the CDW transition under pressure. Recent investigations have revealed that the two *T*_c_(*P*) extrema at *P*_c1_ ≈ 0.7 GPa and *P*_c2_ ≈ 2.0 GPa correspond well to the modification and disappearance of the CDW orders, respectively [12,26].

Following extensive studies on *A*V_3_Sb_5_, *R*V_6_Sn_6_ (*R* = Sc, Y, Gd-Tm) crystallized in the HfFe_6_Ge_6_ structural prototype has been recently discovered as a new family of V-based kagome metals. The structure, electronic and magnetic properties of *R*V_6_Sn_6_ have been studied systematically [33,34,35,36]. As shown in the inset of Figure 1, the V atoms form a perfect kagome lattice while *R* atoms form a triangular lattice; these two sublattices are stacked along the *c* axis in the *ABA* sequence to form a hexagonal layered structure. Recent studies have identified 2D kagome surface states in HoV_6_Sn_6_ and GdV_6_Sn_6_ [36]. Compared with the *A*V_3_Sb_5_ family, the magnetic frustration and anisotropy contributed by the rare-earth *R*^3+^ ions play an important role in determining the low-temperature transport properties of some *R*V_6_Sn_6_ samples [34]. On the other hand, no SC was observed at 0.4 K for *R*V_6_Sn_6_ (*R* = Tb-Tm) [34], 80 mK for *R* = Sc [37], and 2 K for *R* = Y, Gd [33]. In addition, CDW transition is not a prevailing phenomenon in the series of *R*V_6_Sn_6_; however, it is observed in ScV_6_Sn_6_ at *T** ≈ 92 K [37], showing a distinct triple modulation of the unit cell in comparison to the quadruple one observed in *A*V_3_Sb_5_. These observations highlight some distinct characteristics of these two families of kagome metals and thus deserve further investigation. The intimate relationship between CDW and SC in *A*V_3_Sb_5_ and other low-dimensional CDW materials [38,39], raises questions on how the CDW evolves under pressure and whether SC can be induced in the pressurized ScV_6_Sn_6_.

The purpose of this work is thus to address these issues by performing a comprehensive high-pressure transport study on ScV_6_Sn_6_ single crystals under various hydrostatic pressures up to 11 GPa. We find that its CDW transition can be readily destabilized by applying moderate pressure, but no SC was detected down to 40 mK at 2.35 GPa, where the CDW almost vanishes. Our detailed study on the resistivity anomaly around the CDW transition also reveals a subtle modification of the CDW order around 2.04 GPa without signatures of structural phase transition before it vanishes completely. A comparison with *A*V_3_Sb_5_ highlights some distinct characteristics of ScV_6_Sn_6_.

## 2. Materials and Methods

Single crystals of ScV_6_Sn_6_ were grown by using the self-flux method. Sc (99.9%, Aladdin, Shanghai, China) blocks, V (99.9%, Aladdin, Shanghai, China) powder, and Sn (99.999%, Aladdin, Shanghai, China) granules were mixed at a molar ratio of 1:6:40 and placed into an alumina crucible. The crucible was sealed into a quartz tube in a vacuum and was subsequently heated in a furnace to 1150 °C for 15 h. After running the reaction at this temperature for 10 h, the assembly was cooled down to 750 °C within 150 h. The excess Sn was quickly removed at this temperature in a high-speed centrifuge (BIORIDGDE, Shanghai, China), and black crystals with a shining surface and a typical size of 1.5 × 1.5 × 0.5 mm^3^ were left in the crucible. 

Powder XRD measurements were performed on carefully pulverized single-crystal samples using a diffractometer (HUBER, Berching, Germany) at room temperature. Rietveld refinements were performed with the FULLPROF software package (Available online: https://www.ill.eu/sites/fullprof/, accessed on 22 July 2022). Resistivity measurements at ambient pressure (AP) were performed with the Physical Property Measurement System (PPMS-9T, Quantum Design, Beijing, China). In the low-pressure range of up to 2.35 GPa, temperature-dependent resistivity, *ρ*(*T*), was measured with a self-clamped piston-cylinder cell (PCC) while employing Daphne 7373 (Idemitsu Kosan, Tokyo, Japan) as the pressure transmitting medium (PTM). The pressure in PCC was determined from the relative shift of the superconducting transition of Pb according to the following equation: *P* (GPa) = (*T*_0_ − *T*_c_)/0.365, where *T*_0_ = 7.20 K is the *T*_c_ of Pb at AP. In the high-pressure range up to 11 GPa, *ρ*(*T*) was measured with a palm-type cubic anvil cell (CAC, Institute of Physics, Chinese Academy of Sciences, Beijing, China) apparatus. In this case, glycerol was employed as PTM. The pressure in CAC was estimated from the calibration curve at room temperature established by observing the phase transitions of Bi (2.55, 2.7, 7.7 GPa), Sn (9.4 GPa), and Pb (13.4 GPa). Our previous work provides details about the sample assembly and pressure calibrations in the palm CAC [40]. Resistivity measurements at 2.35 GPa were also performed down to 40 mK in a dilution refrigerator (Oxford Instruments, Abingdon, UK).

## 3. Results

Figure 1a shows the room-temperature powder XRD of ScV_6_Sn_6_ after the Rietveld refinement. It confirms that the as-obtained sample is nearly a single phase with several weak peaks from the residual Sn flux indicated by the asterisks. The refinement converged well with small reliability factors, i.e., *R_p_* = 2.42%, *R_wp_* = 3.38%, and *χ^2^* = 2.02. The space group (*P*6/*mmm*) and lattice parameters were also obtained from the Rietveld refinement using the FULLPROF software package. The obtained lattice parameters *a* = 5.4729 Å and *c* = 9.1726 Å are close to those reported in the previous study, i.e., *a* = 5.47497 Å and *c* = 9.17660 Å [37]. 

The ScV_6_Sn_6_ single crystals were then characterized at AP via temperature-dependent resistivity and magnetic susceptibility measurements. Figure 1b shows the *ρ*(*T*) (left red axis guided by the red arrow) curve recorded during cooling from 300 to 2 K at zero field with a current applied in the *ab*-plane. The metallic behavior is confirmed, and a sudden drop of resistivity corresponding to the CDW transition at about 90 K is clearly observed. No SC is detected down to 2 K. We obtained a residual resistivity ratio *RRR* ≡ *ρ*(300 K)/*ρ*(2 K) = 4.66, which is smaller than that of 5.86 reported in the previous study [37]. As shown in the figure, the CDW transition temperature, *T**, can be determined precisely from the sharp peak of d*ρ*/d*T* (right blue axis guided by the blue arrow). Furthermore, the obtained *T** = 88 K is lower than the 92 K reported in Ref. [37], presumably due to the different sample quality as indicated by the smaller *RRR* value. Usually, the formation of a conventional CDW in low-dimensional metals produces a hump-like anomaly in *ρ*(*T*) due to the partial reduction in the density of states at the Fermi level associated with the periodic lattice modulations [41].

In contrast, the observed abrupt drop of *ρ*(*T*) near *T** in ScV_6_Sn_6_ signals a significant modification of the electronic structures accompanying the triple modulation of the unit cell, as suggested in Ref. [37]. As illustrated in Figure 1b, *ρ*_1_ and *ρ*_2_ correspond to the resistivity values at the start and end points of the CDW transition, which are defined by the two intersections of the black lines. To quantify the impact of CDW on the electronic structures of ScV_6_Sn_6_, we define a parameter *α* ≡ (*ρ*_1_ − *ρ*_2_)/*ρ*_1_ to characterize the relative change of resistivity drop at around *T**. A high value of *α* = 0.33 was obtained at AP, indicating a strong electronic structure modification. All these characterizations are consistent with the previous report [37] and further ensure the quality of the studied sample in the present work. 

Figure 2a shows the *ρ*(*T*) curves of ScV_6_Sn_6_ at AP and various hydrostatic pressures up to 2.35 GPa measured in a PCC. The evolution of the CDW transition upon compression can be tracked precisely from the anomaly in *ρ*(*T*) at each pressure. As can be seen, the CDW transition temperature *T** first increases slightly from 88 K under AP to 91 K under the first pressure of 0.43 GPa, and then decreases continuously to lower temperatures with pressure increasing to 2.04 GPa. The CDW transition temperature *T** at each pressure can be determined from the sharp peak of d*ρ*/d*T* as shown in Figure 2b. However, the magnitude of the resistivity drop and the corresponding peak in d*ρ*/d*T* around *T** reduce gradually. At 2.04 GPa, *T** reaches about 34 K, and the parameter *α* decreases to 0.13, signaling the monotonic suppression of CDW by pressure. It is noted that *T** experiences an initial enhancement while the resistivity drop around *T** is weakened during the initial compression process. We attributed these unusual observations to the presence of some strain/stress conditions associated with the solidification of liquid PTM upon cooling down in PCC. Considering the present sample ScV_6_Sn_6_ with a layered structure, the anisotropic stress/strain would induce an initial non-monotonic evolution of *T**(*P*) associated with a slight enhancement of electron–phonon coupling along some peculiar direction.

On the other hand, the presence of stress/strain will influence the transport property by reducing the carrier mobility and continually diminish the transport anomalies and the *RRR* monotonically. Indeed, both the drop of resistivity around *T** and the *RRR* reduce discontinuously at the first pressure of 0.43 GPa compared to those at AP measured outside of PCC, as shown in Figure 2. Since the stress/strain conditions are expected to vary slightly upon further applying pressures, the effect of uniform compression will dominate the pressure effect on the transport properties shown in Figure 2.

Interestingly, at pressures above 2.04 GPa, the abrupt drop of *ρ*(*T*) around *T** disappears and is replaced by a distinct upward bend or shoulder-like anomaly below the characteristic temperature, denoted as *T*′, in order to distinguish it from *T**. This can be seen more clearly from the zoom-in plot in Figure 2c. Accordingly, the anomaly in d*ρ*/d*T* also changes from a sharp peak at *P* ≤ 2.04 GPa to a broad dip at 2.19 ≤ *P* ≤ 2.35 GPa, as seen in Figure 2b. These distinct features of *ρ*(*T*) imply that the CDW order undergoes a significant modification at about 2.1 GPa, which influences the electrical transport properties of ScV_6_Sn_6_. Pressure-induced modification of CDW has also been observed in CsV_3_Sb_5_ and RbV_3_Sb_5_ [12,25,26]. It is noted that the shoulder- or hump-like feature in *ρ*(*T*) around *T*′ in the pressure range 2.19 ≤ *P* ≤ 2.35 GPa is consistent with the conventional CDW as mentioned above. This shoulder-like feature at *T*′ is further continuously suppressed by pressure and becomes a weak one centered around *T*′ = 10 K at 2.35 GPa, implying that the CDW is almost collapsed. 

Figure 3a shows the pressure dependences of *T** (red) and *T*′ (blue) determined from the above *ρ*(*T*) measurements. Although distinct behaviors of *ρ*(*T*) were observed in the vicinity of *T** and *T*′, they show a smooth evolution as a function of pressure. With increasing pressure, *T**(*P*) first increases slightly from 88 K at AP to 91 K at 0.43 GPa and then decreases monotonically to 34 K at 2.04 GPa, above which it connects smoothly to *T*′(*P*). An interpolation of *T*′(*P*) to a zero temperature would predict a critical pressure of 2.4 GPa for the complete suppression of CDW. The evolution of the CDW order can also be viewed from its impact on *ρ*(*T*). As shown in Figure 3b, the pressure dependence of the parameter *α*(*P*) first decreases slowly from 0.33 at AP to 0.30 at 1.15 GPa and then decreases at a larger slope to 0.13 at 2.04 GPa, above which *α* cannot be defined any more. These results indicate that the first-order nature of the CDW transition manifested by the abrupt drop in *ρ*(*T*) is weakened significantly in the pressure range of 1.15 ≤ *P* ≤ 2.04 GPa, and it should be changed to a continuous transition upon compression to *P* > 2.04 GPa. As such, a putative quantum critical point associated with the disruption of the CDW order at zero temperature can be achieved near 2.4 GPa. 

Since CDW and SC are collective electronic phenomena involving strong electron–phonon coupling and Fermi surface instabilities, it is frequently observed that the destabilization of CDW leads to the emergence or enhancement of SC in many systems [12,41,42,43,44]. In the above measurements, we did not observe SC down to 1.5 K at pressures up to 2.35 GPa. To check whether SC would emerge at much lower temperatures, we loaded the PCC at 2.35 GPa in a dilution refrigerator and measured *ρ*(*T*) down to 40 mK. As shown in Figure 2d, no sign of the resistivity drop is observed even at such low temperatures, thus excluding the occurrence of SC in the vicinity of the CDW quantum critical point. 

We further measured the *ρ*(*T*) of another ScV_6_Sn_6_ sample by using CAC in an extended pressure range from 2 to 11 GPa to check whether SC would emerge or structural phase transition would occur at higher pressures. As shown in Figure 4a,b, all *ρ*(*T*) curves show metallic behavior and exhibit smooth variations with temperature/pressure, consistent with the disappearance of CDW order and the absence of any temperature/pressure-driven phase transitions. Still, no SC is observed down to 1.5 K and up to 11 GPa. Instead, a weak upturn in *ρ*(*T*) appears at low temperatures, and the minimum temperature *T*_min_ moves to higher temperatures gradually with increasing pressure, as seen clearly in Figure 4a. The origin of the resistivity upturn is unclear at present; whether it is associated with an intrinsic electronic transition or weak charge localization due to the presence of stress/strain deserves further investigation.

## 4. Discussion

Based on the above transport measurements on ScV_6_Sn_6_ single crystals, we can conclude that the application of high pressure destabilizes the first-order CDW transition and seems to modify it to a conventional CDW above 2.04 GPa until it is entirely suppressed at about 2.4 GPa. No SC is observed down to 40 mK at 2.35 GPa. The smooth variations of resistivity as a function of pressure rule out the possibility of structural phase transition in the studied pressure range of up to 11 GPa. In this section, we briefly discuss these results via side-by-side comparisons with those of *A*V_3_Sb_5_ to highlight the distinct properties of these two vanadium-based kagome metals.

Firstly, for both ScV_6_Sn_6_ and *A*V_3_Sb_5_, irrespective of their distinct forms of CDW, as mentioned above, high pressure effectively suppresses the CDW transition. This trend is commonly observed in many other CDW materials, such as the transition-metal dichalcogenides [45,46,47]. The effect of pressure on CDW can be generally understood by considering the impacts of lattice compression, which reduce the charge modulations and weaken the Fermi surface nesting by shortening the interatomic distances and bandwidth broadening. In addition, we found that the CDW of both systems undergoes some modifications upon compression before it vanishes completely, as reflected by the change of resistivity anomaly around the transition. For example, the occurrence of CDW modification in CsV_3_Sb_5_ is manifested as a change of *ρ*(*T*) around *T** from a kink-like to a hump-like anomaly around 0.6–0.9 GPa [12,26]. In the present case, the sudden drop of *ρ*(*T*) around *T** at lower pressures also transforms into a hump-like anomaly at *P* > 2.04 GPa. Microscopic studies are needed to determine the exact form of the CDW order in ScV_6_Sn_6_ at higher pressures.

Secondly, the consequence of CDW disruption by pressure and its interplay with SC differ in these two systems. At AP, all members of the *A*V_3_Sb_5_ family show the coexistence of CDW and SC [8,11,14,21,48]. In contrast, the *R*V_6_Sn_6_ family with double vanadium kagome layers in the unit cell is not prone to the occurrence of CDW and SC; ScV_6_Sn_6_ is the only exception exhibiting a first-order CDW transition at ~90 K, yet it still shows no SC down to 80 mK [37]. Under high pressure, the coexistent CDW and SC in *A*V_3_Sb_5_ display an intimate relationship, as illustrated by the unusual M-shaped double superconducting dome closely connected with the modification of CDW order [12,25,26]. A considerable enhancement of *T*_c_ is also achieved in *A*V_3_Sb_5_, accompanying the elimination of CDW order [12,25]. However, no SC is observed in ScV_6_Sn_6_ up to 11 GPa. These results indicate that the SC is not a natural consequence of CDW disruption in ScV_6_Sn_6_. 

Although many factors govern the strength of Cooper pairing, the observed distinct interplays between CDW and SC in these two kagome metal systems may be linked to the different CDW orders. According to Ref. [37], the lattice modulations associated with CDW in ScV_6_Sn_6_ and *A*V_3_Sb_5_ show distinct characteristics. The CDW in ScV_6_Sn_6_ is found to exhibit a [1/3 1/3 1/3] wave vector, which triples both the *ab*-plane area and the *c*-axis of the unit cell. The structural refinements suggested that the Sc and Sn1 atoms display the largest modulated displacement up to 0.2 Å along the *c*-axis. In contrast, the vanadium atoms in the kagome layer have a dominant out-of-plane CDW but a much weaker displacement of 0.004–0.023 Å. In addition to the different wave vectors of [1/2 1/2 1/2] or [1/2 1/2 1/4] in *A*V_3_Sb_5_, the vanadium atoms in the kagome layer dominate the CDW transition and display larger in-plane displacements of 0.009–0.085 Å forming either the star of David or tri-hexagonal arrangements [17,21,22,27,49,50,51]. In short, the CDW in ScV_6_Sn_6_ is dominated by the out-of-plane displacements of Sc and Sn1 atoms, while the CDW of *A*V_3_Sb_5_ is governed by the in-plane modulations of vanadium atoms. These distinct characteristics of CDWs should be responsible for the observed difference in consequences of CDW disruption in these two systems. 

Lastly, it is interesting to note that pressure-induced SC has been detected in *A*V_6_Sb_6_ (*A* = K, Rb, Cs), another new class of vanadium-based kagome metals, at pressures above 21 GPa [52]. For this “166” system, no CDW was observed at AP, and the arrangement of vanadium kagome layers is different. Whether ScV_6_Sn_6_ can also become superconducting at pressures of higher than 11 GPa deserves further study. 

## 5. Conclusions

In summary, we performed a high-pressure transport study on the newly discovered kagome metal ScV_6_Sn_6_. We found that CDW transition can be suppressed by pressure, as commonly observed in many other CDW materials. It undergoes some modification at about 2 GPa before it vanishes entirely at about 2.4 GPa. Unlike *A*V_3_Sb_5_, no SC is observed down to 40 mK at 2.35 GPa accompanying the disruption of CDW and down to 1.5 K up to 11 GPa in ScV_6_Sn_6_. Our results indicate that the distinct characteristics of CDW play an essential role in determining the ground state and its response to pressure in vanadium-based kagome metals.

## Figures and Tables

**Figure 1 materials-15-07372-f001:**
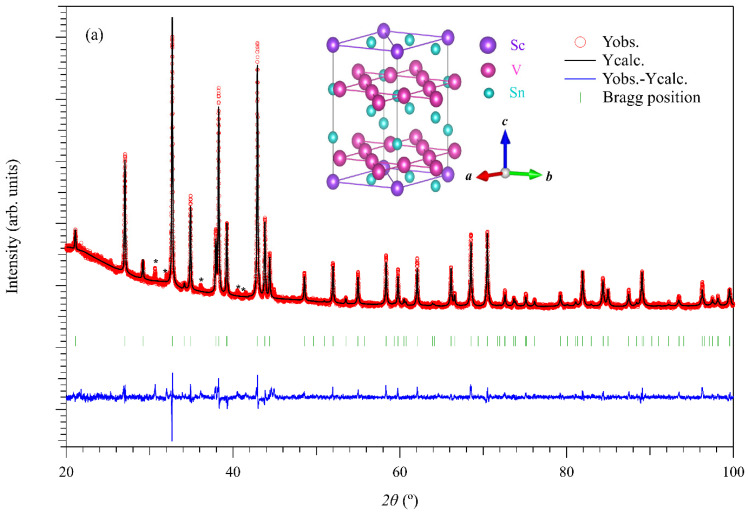
(**a**) Observed (red circle), calculated (black line), and difference (blue line) XRD profiles of ScV_6_Sn_6_ after Rietveld refinements. Inset of (**a**) shows the crystal structure of ScV_6_Sn_6_. (**b**) Temperature dependence of in-plane resistivity *ρ*(*T*) and its derivative d*ρ*/d*T* of ScV_6_Sn_6_ single crystal from 2 to 300 K at ambient pressure.

**Figure 2 materials-15-07372-f002:**
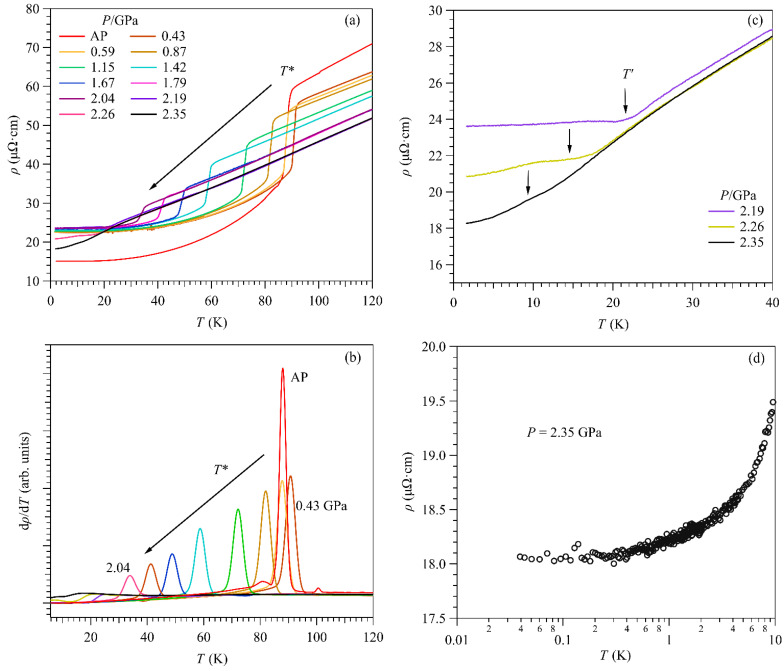
(**a**,**b**) Temperature dependence of resistivity *ρ*(*T*) and its derivative d*ρ*/d*T* for ScV_6_Sn_6_ measured in the piston-cylinder cell under various pressures up to 2.35 GPa. The CDW transition temperature *T** is determined from the peak of d*ρ*/d*T* as shown in (**b**). (**c**) Magnification of low-temperature *ρ*(*T*) data in the pressure range 2.19–2.35 GPa. (**d**) Resistivity down to 40 mK at 2.35 GPa.

**Figure 3 materials-15-07372-f003:**
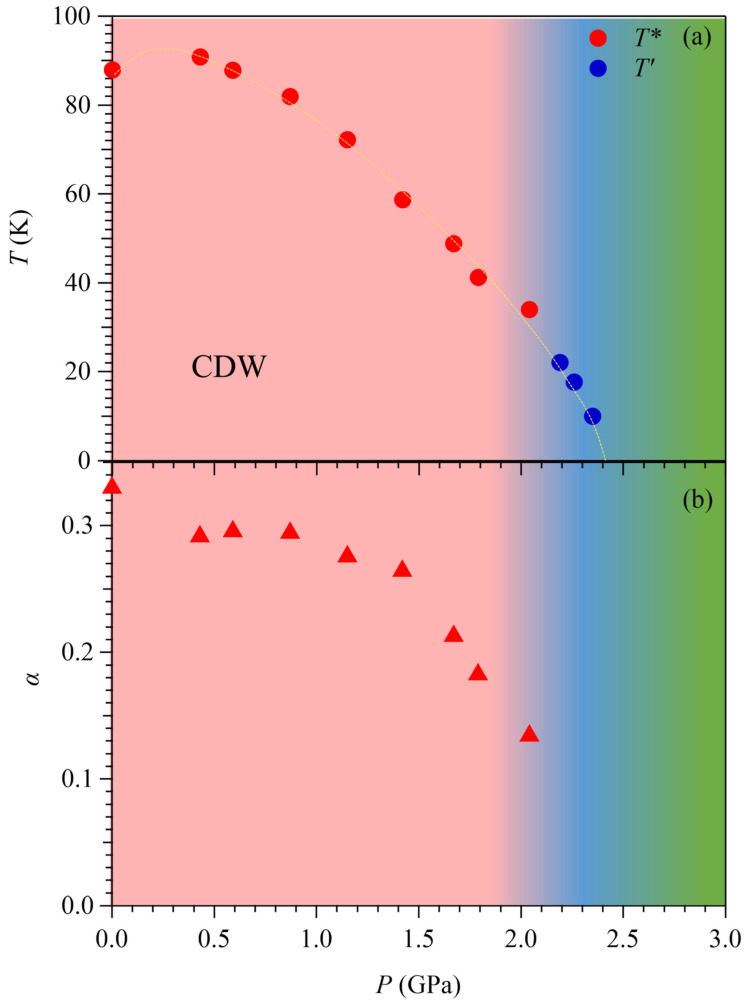
(**a**) Pressure dependences of *T** (red) and *T*′ (blue) for ScV_6_Sn_6_ as determined from the high-pressure resistivity measurements. (**b**) Pressure dependences of the parameter *α*.

**Figure 4 materials-15-07372-f004:**
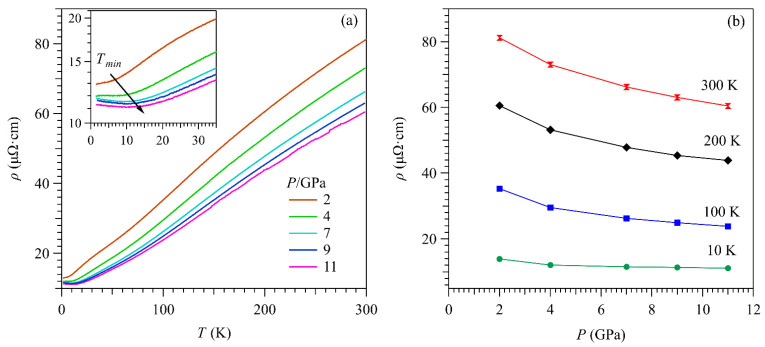
(**a**) Temperature dependence of resistivity for ScV_6_Sn_6_ measured in the cubic anvil cell under various pressures up to 11 GPa. Inset shows the low-temperature data from 1.5 to 35 K. (**b**) Pressure dependence of resistivity at 300, 200, 100, and 10 K.

## Data Availability

Not applicable.

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
