# Peer review of "Destabilization of the Charge Density Wave and the Absence of Superconductivity in ScV6Sn6 under High Pressures up to 11 GPa"

_materials, 2022, doi:10.3390/ma15207372_

Round 1

Reviewer 1 Report

The numbered manuscript and entitled as  Manuscript ID: materials-1939814 Destabilization of the charge density wave and the absence of superconductivity in ScV6Sn6 under high pressures up to 11 GPa, respectively has proved that  a distinct relationship between CDW and SC in ScV6Sn6 in comparison with the well-studied AV3Sb5.

The study is understandable interms of exlanation and what the authors wants to  present. However there are something needd to redesigned such as plots and figures. They are not readeble in their current form. Such as Figure 1 . The other figures are also redesign by correcting the subtitles according to journal style.

In the text the blanck between the words need to be removed

Author Response

Dear Reviewer,

Thank you so much for reviewing our manuscript (materials-1939814) entitled “Destabilization of the charge density wave and the absence of superconductivity in ScV6Sn6 under high pressures up to 11 GPa”. We have revised the manuscript accordingly by fully considering your suggestions and comments. All changes are shown by the “trace back” in the revised manuscript.

Report of the Reviewer 1 -- materials-1939814

The numbered manuscript and entitled as Manuscript ID: materials-1939814 Destabilization of the charge density wave and the absence of superconductivity in ScV6Sn6 under high pressures up to 11 GPa, respectively has proved that a distinct relationship between CDW and SC in ScV6Sn6 in comparison with the well-studied AV3Sb5.

The study is understandable in terms of explanation and what the authors want to present. However, there are something needed to redesigned such as plots and figures. They are not readable in their current form. Such as Figure 1. The other figures are also redesign by correcting the subtitles according to journal style.

In the text the blanck between the words need to be removed

Reply: We appreciated your careful review and valuable comments on our work. We have changed Figures 1 and 2, and removed the blanks between words according to your suggestions.

Reviewer 2 Report

The paper reports the high pressure transport measurement of the Kagome compound ScV6Sn6 of the RV6Sn6 (R = Y, Sc, or Rare earth) family.  Recently, the Kagome lattice structures have recently attracted considerable attention to the scientific community. This is a clear paper reporting a high pressure study of potentially interesting material with Kagome structure.  

The main result indicates that the charge density wave (CDW) transition is suppressed completely by the pressure of about 2.4 GPa but no appearance of superconductivity is detected down to 0.04 K (at 2.35 GPa) or 1.5 K (up to 11 GPa). Finally, the authors compared the results of the ScV6Sn6 compound with well-studied AV3Sb5 compounds. The manuscript follows the conventions of the field, the measurements are professionally executed and described, and the interpretation appears sound. As a result, the paper is probably suitable for publication in something like its current form. I would ask that the authors consider the following comments: 

  • Can the authors comment on why T* first increases a little bit with the application of pressure and then decreases monotonically to 34 K at 2.04 GPa? However, \alpha decreases continuously with the increase of pressure. 

  • What is the physical significance of \alpha. Is there any physical significance of the high value of \alpha at ambient pressure? What is the value of this parameter in other reported isostructural compounds? 

  • What is the value of the other refinement parameters e.g. R, w obtained from SXRD refinement? 

  • There is no axis (and value) in Fig. 1(b). 4.66 is not slightly smaller than 5.86. Please remove the word ‘slightly’ from the text. 

  • The image quality of Figures 1 and 2 is poor specially Fig. 1 (a). Please change it.

  • Include the \rho-T data of ambient pressure in 2 (a) and (b). Then it will be easy to compare the data. 

  • The solid black line in Fig. 1 (b) indicates what? Is it a guide to the eye…Then please mention it clearly.  

  • Some minor typos and grammatical mistakes are there please correct it 

---[…] are subscript in some references e.g. [12,26,27],  

---Follow the same reference format 

Author Response

Dear Reviewer,

Thank you so much for reviewing our manuscript (materials-1939814) entitled “Destabilization of the charge density wave and the absence of superconductivity in ScV6Sn6 under high pressures up to 11 GPa”. We have revised the manuscript accordingly by fully considering your suggestions and comments. Please find the point-by-point response in the file. All changes are shown by the “trace back” in the revised manuscript.

Reviewer 3 Report

The paper presents an experimental study of the ScV6Sn6 superconductor under noticiably high pressures up to 11 GPa, and very low temperatures (down to 4mK in intermediate, but also large, pressures). The results focus on the existence of, and some possible effects induced by, charge density waves. 

This is a topic that is receiving much attention in the last years, being fully current and interesting today. 

The paper is well written. The experimental work is of significant merit, and it seems to have been performed quite professionally. I do not detect any major issue in the research, nor in its presentation. 

Thus, I recommend acceptance of the paper. I only found very minor presentation issues related to the sizes of various elements of the figures (fonts, sizes of axes...). I also did not find an explanation of the meaning of the colors in Fig.3. I recommend that the authors and publisher's services check whether to improve these figures, but from a scientific perspective the paper is fine and from my part I approve its publication in your journal.

Author Response

Dear Reviewer,

Thank you so much for reviewing our manuscript (materials-1939814) entitled ‘Destabilization of the charge density wave and the absence of superconductivity in ScV6Sn6 under high pressures up to 11 GPa’. We have revised the manuscript accordingly by fully considering your suggestions and comments. All changes are shown by the “trace back” in the revised manuscript.

Report of the Reviewer 3 -- materials-1939814

The paper presents an experimental study of the ScV6Sn6 superconductor under noticiably high pressures up to 11 GPa, and very low temperatures (down to 4mK in intermediate, but also large, pressures). The results focus on the existence of, and some possible effects induced by, charge density waves.

This is a topic that is receiving much attention in the last years, being fully current and interesting today.

The paper is well written. The experimental work is of significant merit, and it seems to have been performed quite professionally. I do not detect any major issue in the research, nor in its presentation.

Thus, I recommend acceptance of the paper. I only found very minor presentation issues related to the sizes of various elements of the figures (fonts, sizes of axes...). I also did not find an explanation of the meaning of the colors in Fig.3. I recommend that the authors and publisher's services check whether to improve these figures, but from a scientific perspective the paper is fine and from my part I approve its publication in your journal.

Reply: We appreciated your careful review, positive remarks, and valuable comments to our work. The explanation of the meaning of the colors in Figure 3 has been added in the manuscript. We also have revised our manuscript according to your suggestions.